# Pregnancy and Cancer: Cellular Biology and Mechanisms Affecting the Placenta

**DOI:** 10.3390/cancers13071667

**Published:** 2021-04-01

**Authors:** Melina de Moraes Santos Oliveira, Carla de Moraes Salgado, Lais Rosa Viana, Maria Cristina Cintra Gomes-Marcondes

**Affiliations:** Nutrition and Cancer Laboratory, Department of Structural and Functional Biology, Institute of Biology, University of Campinas, Sao Paulo 13083-862, Brazil; melina.oliveira.bio@gmail.com (M.d.M.S.O.); carlamsalgado@gmail.com (C.d.M.S.)

**Keywords:** cancer during pregnancy, placenta, leucine supplementation, cellular biology

## Abstract

**Simple Summary:**

The main point of this review was to describe most of the mechanisms of the biology of trophoblast cells and neoplastic cells, which point out some similarities between them and the way in which both complex metabolic states could interfere with each other. We show the need for more studies about cancer during pregnancy. In addition, the magnitude of how tumour factors can interfere with the course of pregnancy and affect the foetus’s nutrition and health is the most important point that should be studied to better understand and improve treatment for this complex condition. In this context, we have highlighted the importance of maternal nutritional supplementation with leucine, a branched-chain amino acid that improves the placenta’s metabolism and protects the mother and foetus against the harmful effects of cancer during pregnancy.

**Abstract:**

Cancer during pregnancy is rarely studied due to its low incidence (1:1000). However, as a result of different sociocultural and economic changes, women are postponing pregnancy, so the number of pregnant women with cancer has been increasing in recent years. The importance of studying cancer during pregnancy is not only based on maternal and foetal prognosis, but also on the evolutionary mechanisms of the cell biology of trophoblasts and neoplastic cells, which point out similarities between and suggest new fields for the study of cancer. Moreover, the magnitude of how cancer factors can affect trophoblastic cells, and vice versa, in altering the foetus’s nutrition and health is still a subject to be understood. In this context, the objective of this narrative review was to show that some researchers point out the importance of supplementing branched-chain amino acids, especially leucine, in experimental models of pregnancy associated with women with cancer. A leucine-rich diet may be an interesting strategy to preserve physiological placenta metabolism for protecting the mother and foetus from the harmful effects of cancer during pregnancy.

## 1. Introduction

Cancer can occur in association with other pathologies and physiological states, such as obesity, ageing, and pregnancy [1,2,3]. Cancer associated with pregnancy occurs in one per 1000–2000 pregnancies [4,5,6], and the most common types are breast cancer, melanoma, cervical cancer, lymphoma, and acute leukaemia [7,8,9]. Cancer during pregnancy is considered to occur when a diagnosis is made during the course of pregnancy or up to one year after delivery [10]. As women are postponing pregnancy, the incidence of cancer during pregnancy is increasing [3].

This complex condition results in a worse prognosis and needs more investigation to provide better treatment for both patients, hopefully treating the mother without harming the foetus heath [2,5]. During the first trimester of pregnancy, chemotherapy is not recommended due to a high risk for congenital malformation [10]. Therefore, some chemotherapeutic drugs during the second or third trimester of pregnancy are possible and highly recommended, as long as the mother and foetus are monitored throughout treatment by a highly specialised multidisciplinary team [6,11,12,13].

Since tumour cells and trophoblastic cells share similarities in the biological processes [14,15], there is strong evidence that tumour-induced effects could affect the placental function [16]. The placenta is the centre of pregnancy control, secreting hormones and managing all metabolic processes and functional adjustments. Furthermore, it is necessary to promote blood flow and provide adequate nutrients, gas exchange, and excretion of the foetus [17]. Failures in placenta formation can promote damage and impairment of the foetus and embryo development, as observed in some pathologies such as hypertension, metabolic syndrome, obesity, preeclampsia, and cancer [18,19,20]. Moreover, the placenta contributes to foetal programming, which can be responsive to alterations in the placenta. This can result in adulthood outcomes such as impairment in the synthesis of hormones, and consequently, in metabolic and cardiovascular diseases [21].

In this sense, since maternal nutritional support is critical to conceptus development, some evidence has shown that placental damage occurs in cancer during pregnancy [19,22]. However, more studies are needed, especially with regard to nutritional strategies to mitigate the deleterious tumour effects on the placenta, as well as how a leucine-rich diet has shown promising results in experimental models for both the mother and foetus. [23,24,25]. Therefore, the present article aimed to describe a narrative review on the relationship of placental effects in cancer cells, and vice versa, as well as how leucine participates in modulating their interaction.

## 2. Placenta and Cancer

### 2.1. Placental Physiology

The placenta is a highly active transient organ that is formed by blastocyst invasion into the uterine endometrium. It forms in a controlled process of cellular proliferation and differentiation of placental trophoblastic cells [23,26]. Syncytialisation is among the different types of functions of trophoblasts and cytotrophoblast cells. This process is the most critical event for haemochorial placentation, which is characteristic of rodents and humans. During the placenta formation, cytotrophoblast cells fuse and differentiate into syncytiotrophoblasts (STBs) [27], which are the interface between the mother and foetus.

The placenta has many essential functions, such as foetal nourishment, protection against the maternal immune system, and important endocrine functions that sustain the pregnancy [22,28,29]. The placenta tissue secretes main hormones such as chorionic gonadotropin (hCG), oestrogen, progesterone, leptin, placental lactogen, and placental alkaline phosphatase, as well as some growth factors that control the course of the pregnancy [29]. The placenta cells can also be regulated by glucocorticoids, insulin, and growth factors, resulting in modulation of the placenta activity [30,31].

The STB cells are able to manage and uptake nutrients like glucose, amino acids, and fatty acids provided by the mother. These nutrients are used for the cells’ own metabolism and transport, or they are metabolised for the foetus [28]. Between 70–75% of the O_2_ consumed by the placenta generates adenosine triphosphate (ATP) by mitochondrial oxidative phosphorylation using various substrates, including carbohydrates, amino acids, and probably certain volatile fatty acids. Another interesting point is that the placenta’s glucose consumption corresponds to up to 80% of the glucose provided by the mother’s blood, while the rapid foetal growth is only supported by 20%. In placenta tissue, a key protein called mammalian target of rapamycin (mTOR) is highly activated for not only protein synthesis and cell division, but also the management of the transport and metabolic pathways of the amino acids to support the foetus development, in addition to producing and converting amino acids for its own activity [32]. During placenta formation, with trophoblastic cell invasion, the maternal endometrium arteries are rearranged, forming new blood vessels. This creates a vascular system that can supply the embryo with nutrients, provide gas exchange, and enable foetal excretion [33].

Many hormones are crucial to placenta formation, such as oestrogens, which are involved in trophoblast invasion, development processes, and the stimuli of angiogenesis [34]. Progesterone is an essential hormone for blastocyst implantation, which is essential for the maternal immune response and the reduction of uterine contractility [35]. hCG is extremely important in implantation and nidation and is also related to angiogenesis and the suppression of the myometrium contractions, thus maintaining the pregnancy [36]. This glycoprotein hormone is responsible for development and seems to have evolutionary similarities to some pathways found in cancer cells [37]. In the same evolutionary context, hCG participates in promoting communication between the mother and foetus by enabling extremely invasive placentation [38].

### 2.2. Similarities between Cancer and Placenta Development

Cancer and placenta cells share many similarities regarding growth, multiplication, death resistance, immune modulation, and invasion, and most of these processes have similar proteins and pathways [14]. In cancer cells, some mechanisms are specific for nutrient supply, energy sources, growth, and proliferation [39]. Similar to placenta physiology, which responds to insulin for growth, cancer cells overexpress the insulin receptor, which is highly responsive and promotes rapid growth. This leads to the activation and alteration of the metabolic pathways of glucose, lipids, and proteins [40].

To support this rapid growth, cancer cells have a unique way of consuming energy. Cancer cells prioritise the consumption of glucose by the glycolysis pathway, producing lactate and two ATP molecules, even in the presence of oxygen. This unique glycolytic metabolism of tumour cells is known as the Warburg effect. In addition to the ineffective process in producing high amounts of ATP, the Warburg effect provides advantages to tumour cells, such as quickly providing large amounts of cellular building blocks, including ribose for nucleotides through the pentose phosphate pathway. The effect also provides environmental acidification, which assists the tumour’s progression [41].

Likewise, glucose and essential and nonessential amino acids are necessary for cancer and placenta cell proliferation. Amino acids are important for their role as a metabolite and structural source for growth and cellular proliferation. The protein metabolism that induces synthesis in cancer cells and cooperates with their growth and proliferation is also essential for placenta cells. In this case, mTOR is important as cancer cells reprogram their metabolism to be sustained even with scarce nutrients or in stressful environments. mTOR is responsible for protein, lipid, and other synthesis pathways and is normally activated by the presence of growth factors or nutrients in the case of the placenta or is overactivated in cancer cells [42].

Regarding the similarities between tumour cells and placenta formation, there is some evidence that the tumorigenesis could be related to the change of characteristics present in mature cells to the embryonic characteristics, which are responsible for modulation of the maternal immune system [43]. Some studies have stated that the tumoural characteristics for forming metastasis are concerned with the same trophoblast invasion pattern during the placentation process. The evidence for this occurring in both metastasis and the placentation processes includes mechanisms related to angiogenesis, extracellular matrix degradation, and the expression of specific adhesion molecules [44]. Concerning this point, some studies have proposed that the mechanisms used by trophoblastic cells during implantation are also used by tumour cells to invade and spread throughout the body [44]. These similar mechanisms are related to integrins, extracellular matrix (ECM), and the matrix metalloproteinases (MMPs, which are endopeptidases that enzymatically digest certain proteins in the extracellular matrix) [43]. The invasive trophoblastic cells give an invasive behaviour to the embryo and overexpress specific integrins such as α_5_β_1_, which is characteristic of a haemochorial placenta [45]. Therefore, some metastatic cells lose some specific integrins related to direct basement membranes, while they overexpress α_5_β_1_ integrin and the α_v_β_3_ receptor [46]. The positive regulation of α_v_β_3_ integrin is associated with the invasive and metastatic phenotype expressed in metastatic melanomas and not expressed in noninvasive melanomas [43]. This altering of the integrin expression mechanism is also responsible for activating the trophoblastic cells’ metalloproteinases. Like trophoblastic cells, tumour cells also express a specific MMP, MMP-7, which is a type of metalloproteinase that degrades the ECM [43]. The change in integrin and MMP release plays an essential role in tissue remodelling of trophoblastic cells and endometrial cells. Studies have supported the notion that ECM changes promoted by the activation of tumour MMP can stimulate cell migration and invasion, leading to metastasis [47].

The ability of placental cells to invade the endometrium depends not only on their invasive behaviour, but also on the responses of the endometrium to suppress this invasion and control the immune system [44]. Therefore, some studies have explained that tumour cells’ ability to invade and spread occurs before the existence of the placenta, since some animals without haemochorial placentation, such as marsupials, already have metastatic cancer [48]. Therefore, it is proposed that the metastatic characteristic of tumour cell metastasis depends not only on the invasive cell behaviour, but also on the body’s ability to oppose metastatic invasion, which is derived from the endometrium.

One study correlated the types of placentation (secondary epitheliochorial and haemochorial) and the incidence of cancer metastasis [44]. It was shown that some species that acquired the secondary characteristic of epithelial placentas (such as cattle and horses) have a minor incidence of metastasis compared to those with haemochorial placentation (such as humans, rodents, cats, and dogs), which show a major incidence of cancer metastasis. Thus, there is positive pleiotropy between a maternal capacity to suppress trophoblast invasion, as occurs in cattle and horses, and a lower incidence of metastatic tumours [44]. Similar molecular mechanisms of cancer and trophoblastic cellular machinery and similar pathways are important for understanding the possible effects that cancer cells could have on placenta cells, and vice versa. Some of these similarities are described in the following subsections.

#### 2.2.1. Hypoxia-Inducible Factor Signalling

A similar process to be highlighted is related to hypoxia-inducible factor (HIF), which is activated and overexpressed in cases of tumours and the uterus environment in the state of hypoxia. However, the main difference is between the integrity and function of regulatory processes, which are usually absent in neoplastic cells [14]. The hypoxic environment found in a tumour environment is essential for its invasiveness [49]. In cancer cells, HIF is not only activated in oxygen shortages, but is also stimulated by other proteins, such as mTOR, which interacts with many receptors of tyrosine kinase, whose ligands are growth factors [50]. Another remarkable characteristic is the degradation or lack of expression of HIF regulators, such as phosphatase and tensin homolog (PTEN), p53, and other antitumoural components that block its activity. The absence of these leads to HIF transcription factors activating many genes that promote changes in the cellular metabolism, interfering in many pathways related to angiogenesis and promoting tumour growth and metastasis [51].

A hypoxic environment with approximately 5% O_2_ is necessary to promote trophoblast invasion and differentiation [52]. In this scenario, the foetus and placenta depend on HIF to respond to a lower O_2_ level in the environment. When hypoxia is recognised, the HIFα subunits are translocated into the nucleus to bind with the aryl hydrocarbon receptor nuclear translocator (ARNT) (the HIFβ subunit), promoting the activation of transcription and regulating genes related to placenta formation [53,54]. It has been demonstrated that impaired development of the placenta in the absence of HIF alpha and beta components results in unviability of the embryo [52].

#### 2.2.2. Placenta-Specific Protein-1 Signalling

Other evidence that points to the similarity between cancer development and placenta formation is the expression of placenta-specific protein-1 (PLAC-1). PLAC-1 is expressed in human and mouse trophoblasts. PLAC-1 has an essential role in the regulation of placenta development, especially in trophoblast growth and differentiation [55], which are responsible for trophoblast cell migration and invasion [56]. It also has a role in trophoblastic syncytialisation [27], which is important for the activity of the placenta. This protein also is responsible for embryo formation since the trophocyte invasion into the endometrium needs an angiogenesis process [55].

PLAC-1 also plays an important role in neuron development, particularly foetal brain development [55,57,58,59,60]. During embryo implantation, the trophocyte invasion into the endometrium and blood vessel formation is influenced by PLAC-1, which is very similar to the growth, invasion, and migration of tumours [56]. Thus, despite trophoblastic cells, PLAC-1 is not found in healthy and differentiated cells except for some cancer cells. In these cases, this protein might help cancer cells to mimic the early stages of placental development. Some trophoblastic characteristics are shown in this case, such as immunomodulation pathways regarding chemokines and immunotolerance, which help tumour growth [61]. PLAC-1 is found in colorectal cancers and non-small cell lung cancer (NSCLC). In these types of tumours, PLAC-1 is directly related to tumorigenesis and aggressiveness, which are involved in the processes of cellular proliferation, migration, and invasion [62]. However, the molecular pathways in which PLAC-1 participates in the trophoblastic invasion and differentiation with similar processes in cancer cells are not totally clear.

#### 2.2.3. Placenta Growth Factor

Another molecular similarity between cancer and the placenta is the placenta growth factor (PlGF), a glycoprotein secreted and found in STBs and decidua cells. PlGF belongs to the family of vascular endothelial growth factors (VEGFs) and increases during all stages throughout the course of pregnancy, except during labour. It acts in several tissues, such as the placenta, heart, vessels, and skeletal muscle. The main action is developing angiogenesis and maintaining the blood vessels through direct action, and indirectly by VEGF [63,64]. Thus, in some cancer cells, PlGF can alter certain tumorigenic characteristics and worsen the prognosis and survival in cancer patients [65]. PlGF helps angiogenesis and is present in tumoural tissues at a higher level compared to healthy tissues. The expression levels of PlGF in tumour tissues compared to nontumour tissues are also higher in many other types of cancer, including breast, bladder, prostate, and colon cancer. This corresponds to a higher degree of lymph node invasion in these patients [65,66].

Cancer and the placenta’s relation is even more complex, since both can affect each other’s functionality. Some pregnancy hormones released by trophoblasts (such as hCG, oestrogen, progesterone, and placental lactogen) can induce cancer onset and promote the progression and malignancy of tumour cells [67,68,69,70]. Some cancers can also promote placental damage, leading to harm to the foetus, such as IUGR and premature births [16,23,24].

#### 2.2.4. Cancer Response Due to Placental Factors

The study of the effects of pregnancy-released products on cancer cells are rare and hardly ever explored. However, some effects can be listed regarding the pregnancy hormones secreted by trophoblasts.

#### 2.2.5. hCG Effects in Cancer Cells

hCG is a hyperglycosylated molecule that is important for the endometrial implantation of trophoblasts and for the course of pregnancy. Some studies have shown that the elevation of hCG in early pregnancy is related to small fibroids, which are precursors of leiomyomas [67]. The possible correlation between hCG and leiomyoma development could be explained by the similarities of the β-chain of hCG to LH, which can stimulate the same receptors [71]. The myoblast LH/hCG receptor expression is increased at a specific time of pregnancy and depressed in the moment of labour [72]. Also, the presence of messenger RNA (mRNA) of these LH/hCG receptors was observed in leiomyomas. hCG seems to increase the number of fibroid cell numbers and stimulate myometrial cell proliferation [67].

hCG is also important for tumours due to the angiogenic and immunogenic function. It is also an important biomarker in cancer [73]. The action of hCG in choriocarcinoma cells promotes their invasion, which characterises a poor prognosis of patients. Nevertheless, some tumour cells, such as bladder cancer cells, can produce hCG, leading to higher progression and malignancy [74]. hCG in testicular cancer is also well known and was detected in the cyst fluid of cyst germinomas, craniopharyngiomas, and pituitary adenomas [68]. This evidence shows that hCG is produced in a healthy state, as in pregnancy, but has a linear relation to tumourigenic activity and aggressiveness, promoting tumour growth when secreted by cancer cells.

#### 2.2.6. Oestrogen, Progesterone, and Other Factors Related to Pregnancy That Affect Cancer Cells

Besides the functions in reproduction, brain function, and bone density, oestrogen is related to various cancer types, especially breast cancer. Additionally, breast cancer can be masked during pregnancy since the mammary gland development is more influenced by enhanced oestrogen levels [75]. The main action of oestrogen is a higher capacity to stimulate cell proliferation and progression [69]. Oestrogen also promotes the expression of *c-myc*, a transcription factor encoded by the proto-oncogene *c-myc*, which is responsible for cell viability and growth [76]. Moreover, oestrogen can stimulate cyclin D1, an important cell regulator responsible for integrating extracellular signals and cell cycle progression, and it becomes an oncogene when deregulated [77]. Oestrogen is also responsible for the inhibition of the upregulation of antiapoptotic proteins Bcl-2 and Bcl-x [69,78]. An oestrogen-responsive cancer can be affected not only by oestrogen, but also by other growth factors present in the physiological adaptation of pregnancy. For example, the transmembrane glycoprotein mucin-1 (MUC-1) present in the maternal–foetal interface is secreted by uterine cells and is responsible for maintaining trophoblastic cells. It is very similar to the MUC-1 produced in breast cancer [79]. Moreover, other studies showed that women diagnosed with breast cancer up to the first five years postpartum were identified to have more aggressive subtypes of tumours, such as luminal B-like and triple-negative breast cancer, and they had worse clinical prognosis [80,81].

In an experimental model, high expression of P63 (a transcript factor homologous to P53) played an important role in increasing the risk of breast cancer. A study in mice showed that p63 induces differentiation and self-renewal in breast epithelial cells that emerge after pregnancy [80,82]. In young pregnant women with breast cancer, the most common type of tumour is invasive ductal carcinoma, which is found in 80–90% of cases and corresponds to a higher prevalence of hormone-receptor (HR)-negative status, human epidermal growth factor receptor 2 (HER2)-positive status, and Ki67-positive status [83,84].

Progesterone has great importance for reproductive physiology and other functions, such as neuroprotection, neurogenesis, and regulation of the immune system [70]. Many studies have reported the preventive function of progesterone in women’s health during pregnancy and in the long term. During pregnancy, placental secretion of oestrogen and progesterone is essential for mammary gland differentiation and growth, acting as protective components against breast cancer [75]. Furthermore, other studies have also demonstrated a protective effect of progesterone in relation to the risk of breast cancer and other types of cancer [82]. There is a positive correlation of higher progesterone levels in early pregnancy with lower risk of ER+/PR+ breast tumours. In addition, higher oestrogen levels and lower progesterone levels are correlated with a higher risk of breast cancer [75,79,85]. Nevertheless, ER+/PR+ cancer cells have some interferences from other hormones like testosterone [85].

In addition to breast cancer, studies have suggested that the expression of ER-α and PR in cancer cells can be used as a guide in the evaluation of the prognosis of diseases, such as colon cancer and lung cancer [86,87]. The lack of ER-β and the presence of ER-α are related to an impairment of the patient outcome and tumour progression, with no relation to the presence of PR [86]. The expression of ER in colon tumour cells is different between pregnant and nonpregnant patients. The nonfunctioning of the P53 protein, relating to the tolerance of the maternal immune system and the role of cyclooxygenase (COX-2), is a contributing factor to the progressing of colon cancer [88]. Most of this correlation comes from findings that point to a higher concentration of COX-2 in colon tumour cells [89]. Another study showed that the lack of ER-α/ER-β [90] and PR is related to the presence of growth factors and prostaglandins from the pregnancy state, which could boost tumour growth. Then, the angiogenic factors released by the placenta, such as TGF-b, can induce an angiogenic process that favours tumour growth. Subsequently, the tumour growth could also be influenced by the immunosuppressive state in pregnancy [91]. These points emphasise the need for more studies on colon cancer biology and the relation of oestrogen and progesterone to patient prognosis [86].

### 2.3. Placental Impairment Due to Cancer Association

Concerning all the similarities between cancer growth and the formation of the placenta, our previous findings have shown impairments of the placenta due to the presence of cancer [16,92,93]. There is no doubt about the importance of placenta formation to the success of foetal growth and development. Therefore, placental impairment could promote foetal growth restriction (FGR) or intrauterine growth restriction (IUGR), which can be associated with cancer, as these two situations require high nutritional support [94,95]. However, most studies evaluating the consequences of cancer during pregnancy are done with an experimental model [16,92,93].

Some of our previous findings have shown impairment of the placenta due to the presence of cancer or some tumour factors, such as proinflammatory cytokines interleukin-6 (IL-6), interferon-gamma (IFN-γ), and tumour-necrosis factor (TNFα) [24]. One of these studies evaluated the consequences of cancer in pregnant rats and showed that placenta protein and total DNA content were decreased, as well as a reduction in the labyrinth zone in the presence of cancer. This shows an impairment of the exchange between the placenta and foetus [16,24,96]. Furthermore, a negative correlation was found between the placental and foetal weight in a tumour-bearing rat group, suggesting an inefficient placenta compared to a non-tumour-bearing rat group [16]. The tumour consequences in this experimental model also showed effects on foetus growth and metabolism, jeopardising the foetal viability and inducing foetal resorption (as in abortion in women). Associated with these damages was a deep reduction in serum protein, albumin, glucose, and skeletal muscle protein synthesis in the foetus as harmful effects of the tumour development [86].

Previous research with an experimental model showed that tumour factors, such as proteolysis-inducing factor (PIF), and pro-inflammatory cytokines, such as IL-6 and TNF-α, could jeopardise the placenta tissue, impairing the connection between the dams and foetus [24]. These cytokines can influence the placenta’s integrity and function, interfere in foetal programming, and lead to DNA damage in hematopoietic cells from the uterus. This could put the child at a high risk of leukaemia [97]. In our previous study, we evaluated a trophoblastic cell line—BeWo, in cultures exposed to PIF; these trophoblastic cells had increased DNA damage and oxidative stress, showing the direct damage effect of tumour growth [32,51].

The correct interactions between immune cells and the placenta are important to guarantee the formation and vascularisation function of the maternal/foetal unit [98,99]. Both placental and cancer development have similar characteristics regarding immune tolerance, which can promote body invasion and proliferation processes. Increased vessel resistance, for example, occurs through unnatural interactions between natural killer cells (NKs), which reside in the decidua layer bordering trophoblasts. This leads to abnormal implantation and the higher release of mast cell granules, which could alter the vascular net in the placenta [100]. In the same way, cancer plays an important modulation role in the immune system, helping neoplastic cells develop and spread to other organs. A low concentration of some cytokines, such as TGF-β, is a factor in pretumour suppression. However, an elevated concentration of TGF-β associated with the anti-inflammatory IL-10 can act as stimulatory factors for tumour proliferation [91]. Interferon-gamma (IFN-γ) has a dual role in tumour progression, inhibiting it when at low levels and promoting at high levels. Meanwhile, IFN-γ and IL-10 are significantly higher in normal pregnancy to protect the trophoblasts from the mother’s immune response. Thus, cancer has some similar growth to foetus development, utilising similar immune system modulation mechanisms for its own growth [101].

In order to avoid the immune system, trophoblast cells express some subtypes of the histocompatibility complex (HLA, such as nonclassical subtypes HLA-C, HLA-E, and HLA-G [102]). These are not recognised by the maternal T cells and lead to success of the pregnancy. Similarly, cancer can avoid attack from the immune cells because these cancer cells lack the expression of HLA, class I, which occurs through a genetic or epigenetic process [103,104]. They show pre-existing self-tolerance pathways [102], which are a common mechanism in the human body to avoid autoimmunity. This autoimmune mechanism depends on when the development process of T cells in the thymus does not recognise self-antigens/MHC complexes through the thymocytes receptors or by peripheric tolerance mechanisms that are related to the lack of T cell proliferation in response to antigen stimulation [105].

In the case of the placenta, the incorrect procedures of some pathways regarding the immune tolerance and the immune system in some placental structural processes can lead to diseases such as intrauterine growth restriction, spontaneous abortion, and congenital infection [106]. Another example is related to melanoma, which can spread metastasis to the placenta. Most of the mothers die at approximately seven months in these cases, which shows the importance of more studies in this area [107]. Metastasis in the foetus is extremely rare, with only six reported cases [108], including melanomas, lung cancer, and lymphomas. These cases occur because of a nonfunctional activity of the placenta and the lack of a foetal immune system, which is mainly related to tumour effects [108,109]. However, the interference of the immunological system changes in the context of pregnancy associated with cancer, and the cancer-cell-induced damage in the placenta needs further studies.

Therefore, considering that cancer promotes damage to the placenta, the importance of studying it resides not only in the mother and foetus’s health, but also in discovering mechanisms that could contribute to new therapies. Although chemotherapy and radiotherapy treatments are considered incompatible with pregnancy, recent studies recommended conventional cancer treatments during pregnancy [6,110]. Surgical interventions can be performed at any time during pregnancy, except genital cancer surgeries, which are still a challenge [6,111]. During the second and third trimesters, the placenta provides protection from the effects of chemotherapy, acting as a barrier. Thus, some chemotherapeutic drugs, such as doxorubicin, cyclophosphamide, and 5-fluorouracil, can be safely administered with no apparent damage to the foetus or newborn [110,112]. However, Berveiller and colleagues demonstrated a low transplacental transfer of the chemotherapeutic drug paclitaxel, but pointed out that a long-term exposure could lead to accumulation of this drug in the placenta and a possible release into the foetal circulation. In addition, the authors showed that the administration of this chemotherapy during pregnancy downregulated the transcription of ABC and SLC transporters in the placenta and significantly increased the expression of ATP7B/WND—placental transporters that regulate the cell efflux of the chemotherapeutic drugs [113]. Moreover, IUGR was related to oxidative damage to trophoblastic cell DNA, such as 8-hydroxy-2′-deoxyguanosine (8-OHdG), in pregnant women exposed to chemotherapy for more than 196 days of gestation [11]. Thus, the apoptosis process in the epithelial layers of amnion and corium trophoblast may be directly facilitated by exposure to chemotherapy [110], which jeopardises the interaction between placenta and foetus. Then, further studies are needed to understand the consequences of chemotherapy on changes in the regulation of these placental transporters, and the consequences for both placenta and foetus. Although it is recommended to initiate cancer treatment during pregnancy, especially for diagnosis of advanced cancer, foetuses exposed to chemo and/or radiotherapy have a high rate of premature birth, which is associated with long-term morbidities and impaired cognition [110,114]. Indeed, the effects of chemotherapy during pregnancy show new areas for further studies on the epigenetic effects in adult offspring [110]. Most of the affected pathways presented in this review are summarised in Table 1.

Other coadjutant therapies are important to administer to patients with cancer to improve their prognosis. In this scenario, nutritional supplementation with leucine could be an interesting approach. With an experimental model of cancer, some preclinical studies have shown some positive effects of a leucine-rich diet on placenta tissue and foetal health [16,23,24,96]. Although preclinical studies show positive results with leucine, to our knowledge, until now, no study has investigated the role of leucine in pregnant patients with cancer.

## 3. Leucine-Rich Diet as a Potential Treatment

Cancer during pregnancy is a condition that requires treatment to improve health for both the mother and foetus. However, to decide which treatment strategies are needed in each case, both of them must be evaluated individually by a multidisciplinary medical team, including obstetricians, gynaecologists, oncologists, paediatricians, and psychologists [116]. Different modalities should be analysed regarding the treatments, such as surgery, chemotherapy, radiological, and nutritional treatments for each patient [117].

Nutritional supplementation with leucine could be beneficial as a co-adjuvant treatment for preserving both the mother and foetus. Leucine is an essential branched-chain amino acid (BCAA), like valine and isoleucine, acting as a signalling molecule that stimulates protein synthesis by activating the mTOR pathway in skeletal muscle, adipose tissue [118], and placenta cells [16,23,24]. The effects of leucine, such as anabolic stimulus, are already known in younger people, adults, and the elderly, in whom it assists the increase in skeletal-muscle protein synthesis [119,120]. However, dietary leucine supplementation studies in the context of cancer during pregnancy need to be enhanced. Most of these studies were preclinical assays.

In the ovine uterus, mTOR pathway stimulation by leucine regulates cell growth, mainly affecting the protein translation in response to the availability of nutrients and growth factors [121]. Thus, leucine guarantees mTOR pathway maintenance, which was essential for cellular growth and proliferation in the uterus and placenta tissue in an experimental model with rats [122]. However, in the pathological state of cancer, changes in mTOR signalling can be harmful, affecting protein translation and cellular metabolism [32,123]. mTORC1 is a protein complex that is activated by the inhibition of the inhibitory complex tuberous sclerosis proteins 1 and 2 (TSC 1/2) through protein kinase B (Akt). It can integrate extracellular signals transduced by kinases in protein synthesis by the activation of ribosomal protein S6 kinase beta-1 (S6K1) and eukaryotic translation initiation factor 4E-binding protein 1 (4EBP1), which leads to protein synthesis [66]. Our previous studies have shown that leucine can modulate the damage effects in cancer, improving the responses of these upstream (Akt) and downstream (S6K1 and 4EBP1) proteins and activating mTOR in placental tissue [16,23,24].

Cancer during pregnancy could lead to a nutritional competition between the foetus and tumour cells, and the damage of the tumoural factors can compromise the placenta, contributing to the impairment of the foetal development [24]. Foetal growth and maternal physiological adaptations during pregnancy require an acceptable amount of nutrients to ensure a healthy pregnancy. In pregnant rats, tumoural factors such as PIF, IL-6, and TNF-α could promote a reduction of amino acid uptake by the placenta, which has been evidenced by low activation of mTOR pathway in this tissue [16,24,32]. As mentioned above, a leucine-rich diet is beneficial to minimise the deleterious effects of lean mass loss induced by the evolution of tumour growth [115]. In addition, in an experimental mouse model of cancer-induced cachexia (murine colon adenocarcinoma (MAC16)) during pregnancy, we also demonstrated that a leucine-rich diet decreased foetal resorption, recovered foetal weight, and modulated mTOR expression in placental cells [24,124]. Our study observed that in pregnant mice treated with PIF obtained from MAC16 tumour, skeletal muscle spoliation occurred, which indicates that tumour factors promote deleterious effects to healthy tissues and foetal tissues [23,24]. Additionally, the leucine-rich diet minimised some deleterious parameters found in tumour-bearing mice injected with tumour factor PIF. This reinforces that this amino acid has a protective effect in the case of unhealthy conditions promoted by tumour development [24].

Moreover, some experiments involving lactating sows demonstrated that nutritional supplementation with BCAAs increased the release of amino acids in milk [125], whereas amino-acid-deficient diets (especially in leucine) decreased milk synthesis through mTOR deactivation. Furthermore, a lack of BCAAs impairs foetal development, jeopardising muscle development since BCAAs are necessary for mTOR activation [126]. Moreover, a diet supplemented with BCAAs, mainly with leucine, may prevent IUGR by activating mTOR via IGF-1/2 signalling in the foetus’s liver [127]. Furthermore, studies in mice have demonstrated the BCAAs’ importance in embryonic implantation, improvement of the blastocyst quality, and trophectoderm motility [128].

Leucine can also interfere with the offspring’s epigenetic programming in rats when the mothers are fed a leucine-rich diet during pregnancy and breastfeeding. The adult offspring of tumour-bearing rats had a reduction in cachectic state, avoided muscle loss and impairment of protein synthesis, and minimised protein degradation. These epigenetic effects were also found in morphometric parameters, such as food intake, triglycerides, and serum levels of albumin and globulin. This led to the maintenance of the gastrocnemius muscle due to the upregulation of the mTOR activity [129,130,131]. Thus, leucine has been demonstrated to be a potential co-adjuvant therapy in animal models. However, more clinical studies are necessary to learn about how leucine could modulate deleterious cancer effects in pregnant women.

## 4. Conclusions

Due to its great magnitude of importance, cancer during pregnancy needs further elucidation regarding the similarities, differences, and also the consequences of the conventional therapy in order to find new drugs and conducts for better treatment of the mother and foetus. As preclinical studies have shown, leucine supplementation could be a co-adjuvant therapy as it effectively maintains the mTOR pathway in animal models, minimising some damage effects in the placenta and foetus in the context of cancer (Figure 1). However, further clinical studies in this area are needed to elucidate how the communication between cancer and physiology in pregnancy works, as well as the effect of leucine on trophoblastic and cancer cells, especially in humans.

## Figures and Tables

**Figure 1 cancers-13-01667-f001:**
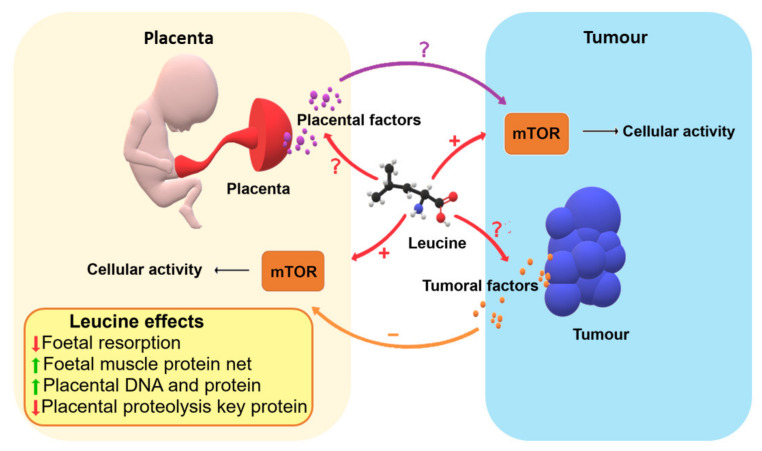
Mechanisms of how leucine nutritional supplementation to the mother could improve some points in minimising cancer damage factors. Foetal resorption could be decreased with a positive foetal muscle protein net, and the placenta may also benefit with a decrease in proteolysis, maintaining the protein synthesis/proteolysis ratio, which leads to physiological DNA and protein levels.

**Table 1 cancers-13-01667-t001:** Molecular similarities between placenta and tumour tissues and molecular pathways and key proteins expression related to the placenta or tumour effects.

Cancer during Pregnancy			
Key Proteins	Similarities		
	Placenta	Tumour	Refs.
MMP7	√	√	[43,47]
Integrin α_5_β_1_	√	√	[45,46]
HIF	√	√	[14,52]
PLAC-1	√	√	[55,56]
PlGF	√	√	[63,65]
Pathways and key proteins	Placenta affecting tumour evolution	Tumour affecting placenta activity	
hCG	↑	unknown	[67,73]
Oestrogen	↑	unknown	[79]
Progesterone	↓	unknown	[70,75]
Proinflammatory cytokines	↑	↑	[24,51,101]
IL-10	unknown	unknown	[101,115]
mTOR	unknown	↓	[16,24,32]
4EBP1	unknown	↑	[16,24]
Degradation Proteins	unknown	↑	[16]

Legend: ↑ up regulated; ↓ down regulated; √ similar key proteins.

## Data Availability

No new data were created or analyzed in this study. Data sharing is not applicable to this article.

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
