# Peer review of "Pregnancy and Cancer: Cellular Biology and Mechanisms Affecting the Placenta"

_cancers, 2021, doi:10.3390/cancers13071667_

Round 1
Reviewer 1 Report
Thank you to the authors for addressing all of my previous comments
Author Response
Again, we would like to thank you for the time spent in analysing our manuscript and for your valuable comments.
Reviewer 2 Report
Thank you for this revised document that is now clearer and very interesting.
Please find some comments that should be taken into account:
L64-65: Please moderate again.
L84: add a reference
L402: ref 12 is not adapted. Please add other references, recent and solid.
L402-407: too short. There are other mechanisms that could be involved in IUGR. There are some studies about placental effects of chemotherapy. Please add information.
In the figure, what are the effects of formation of ribosomal complex. Please complete.
There are a repetition of mTOR pathways on the figure. I do not understand why.
Regarding all molecular mechanisms presented by the authors, I think that a table or a figure summarizing all the data may be very useful (mTOR pathway, MMP, HIF,TNF, IL6, etc…) and lead to a easy-to-read article. The example of the provided figure is good and useful for readers.
Finally, the paragraph regarding the use of leucine diet in highly interesting, but perhaps it takes too much importance given the amount of data we do not have in the field of cancer and pregnancy… We have the feeling that leucine represent the “miracle treatment”. Conversely, perhaps the authors should complete the paragraph regarding the use of chemo during pregnancy and its effects on placenta (short paragraph in the revised manuscript). We know that chemo may induce transporters (Berveiller et al…), may lead to apoptosis (Van Calsteren et al. to my knowledge)…Please complete.
Thank you for this highly interesting job!
Author Response
Campinas, Brazil – March 20th, 2021
Reviewer #2
Thank you for this revised document that is now clearer and very interesting.
Answer: Again, we would like to thank you for your comments, and suggestions for minor review in our manuscript, that we are sure that improved the whole manuscript.
Please find some comments that should be taken into account:
L64-65: Please moderate again.
Answer: We thank you for this advice, and we have now rewritten these lines.
L84: add a reference
Answer: The reference was added in line 85
L402: ref 12 is not adapted. Please add other references, recent and solid.
Answer: Thank you for raising this point, and we have now included a proper reference for this point.
L402-407: too short. There are other mechanisms that could be involved in IUGR. There are some studies about placental effects of chemotherapy. Please add information.
Answer: We have now rewritten this paragraph and included more information about the effects of chemotherapy on placenta and foetus development. We hope this has improved this new version.
In the figure, what are the effects of formation of ribosomal complex. Please complete.
Answer: We modified the figure to better improve the pathways altered by the effects of tumour and placenta, trying to summarise the effects presented in the whole manuscript.
There are a repetition of mTOR pathways on the figure. I do not understand why.
Answer: We modified the figure to better improve the pathways altered.
Regarding all molecular mechanisms presented by the authors, I think that a table or a figure summarizing all the data may be very useful (mTOR pathway, MMP, HIF,TNF, IL6, etc…) and lead to a easy-to-read article. The example of the provided figure is good and useful for readers.
Answer: As mentioned before, we thank you for this suggestion and agree that a table would be very useful, so we have added a Table showing the main effects summarising the data presented in the manuscript.
Finally, the paragraph regarding the use of leucine diet in highly interesting, but perhaps it takes too much importance given the amount of data we do not have in the field of cancer and pregnancy… We have the feeling that leucine represent the “miracle treatment”. Conversely, perhaps the authors should complete the paragraph regarding the use of chemo during pregnancy and its effects on placenta (short paragraph in the revised manuscript). We know that chemo may induce transporters (Berveiller et al…), may lead to apoptosis (Van Calsteren et al. to my knowledge)…Please complete.
Answer: We modified the figure to better improve the pathways altered by the effects of tumour and placenta, trying to summarise the effects presented in the whole manuscript.
Thank you for this highly interesting job!
Answer: We do appreciate your comment and thank you for the time spent in analysing our manuscript and for the valuable comments and suggestions.

This manuscript is a resubmission of an earlier submission. The following is a list of the peer review reports and author responses from that submission.
Round 1
Reviewer 1 Report
This manuscript is a review article discussing the mechanisms active during pregnancy and placentation that are similar to mechanisms occurring during cancer progression and development. The review seeks to elucidate similarities between cancer and cancer development while also discussing the detrimental effects on the placenta that occur during a pregnancy co-existing with cancer.
Comments
- Section 2.1 should be have further subsection headings indicating the mechanisms/pathway that is being described (i.e. Hypoxia inducible factor signaling; Cancer metastasis and trophoblast invasion)
- Lines 99-109 state a number of hypothesis regarding similarities between placental formation and tumor cells. The authors should expand their discussion of these hypotheses and the evidence that supports and/or refutes these hypothesis. Simply stating the hypotheses does not help the audience understand the gaps in knowledge to make decisions on what areas still require further study. For example, in lines 102-103- expand discussion of shared mechanisms occurring in metastasis and placentation. Discuss genes or proteins that are expressed during these processes and their roles in the success of these processes.
- Lines 110-119 What is the role of PLAC-1 in placenta and embryo formation? The authors state 9ow it may help cancer cell by mimicking trophoblast characteristics. For this statement to be relevant, you must introduce what PAC-1 does in placenta and embryo formation to trophoblasts.
- Lines 128-137- Authors need to discuss any specifics known on how the placenta affects cancer progression.
- The entire section 2.1 provides generalized statements and not many very specific pathways, proteins, mechanisms. The statements are more associative and hint at knowledge of specific mechanisms being identified in research. The article should discuss these specifics to give the audience an overview of what is known in the field.
- Please describe the model that is referred to in lines 145-153.
- Lines 175-180 do not seem to be related to the preceding statements in the paragraph. What “factors” are the authors referring to in line 175?
- Lines 197-215 An expanded explanation is need on immune activity in pregnancy and similar mechanisms in cancer progression/development.
Overall the review addresses a topic of importance given the increasing prevelance of cancer diagnosis during pregnancy. The manuscript reads as if one individuals composed sections 1 and 2 with another individual composed the latter 2 sections. This should be corrected so that the review does not seem so disjointed. Section2 of the manuscript should be revised to add in additional specifics of mechanisms and details regarding what is currently known as was done in section 3. Section 2 seems disjointed and jumps from topic to topic without transition statements to connect the paragraphs and is not reflective of the subheading. Additionally, English type editing is required for the first 2 sections of the manuscript.
Author Response
Campinas, Brazil – March 04th, 2021
Reviewer Comments and Author Answers
Reviewer #1:
This manuscript is a review article discussing the mechanisms active during pregnancy and placentation that are similar to mechanisms occurring during cancer progression and development. The review seeks to elucidate similarities between cancer and cancer development while also discussing the detrimental effects on the placenta that occur during a pregnancy co-existing with cancer.
Answer: The authors Thank you for the time spent in analysing our manuscript and for the valuable comments and suggestions. We are completely certain that the revised manuscript has improved substantially thanks to your suggestions. The changes made in the revised paper are highlighted in yellow.
Comment 1. Section 2.1 should behave further subsection headings indicating the mechanisms/pathway that is being described (i.e. Hypoxia-inducible factor signalling; Cancer metastasis and trophoblast invasion).
Answer: Thank you for this comment. We have rewritten the manuscript, and as you suggested, we added additional subsections to better organise all the ideas and point out some molecular mechanisms and pathways.
Comment 2. Lines 99-109 state several hypotheses regarding similarities between placental formation and tumour cells. The authors should expand their discussion of these hypotheses and the evidence that supports and/or refutes these hypotheses. Simply stating the hypotheses does not help the audience understand the gaps in knowledge to make decisions on what areas still require further study. For example, in lines 102-103- expand the discussion of shared mechanisms occurring in metastasis and placentation. Discuss genes or proteins that are expressed during these processes and their roles in the success of these processes.
Answer: Again, thank you for this comment. We have rewritten these statements in order to better explain and exemplify the similarities between cancer and trophoblastic cells. Some studies state that the tumoural characteristics forming metastasis are concerned with the same pattern of trophoblast invasion. Thus, we have rewritten these statements and focused on the mechanisms related to angiogenesis and degradation of the extracellular matrix that are presented in both tumour and trophoblastic cells. This part of the manuscript is now in lines 119 - 138
Comment 3. Lines 110-119 What is the role of PLAC-1 in placenta and embryo formation? The authors state how it may help cancer cell by mimicking trophoblast characteristics. For this statement to be relevant, you must introduce what PAC-1 does in placenta and embryo formation to trophoblasts.
Answer: We agree and have sectioned the manuscript. We have rewritten all the points following the suggestion, introduced the physiology of the placenta, and discuss what is altered and worsened in cancer cells. Changes were done for this specific topic to elucidate the role of PLAC-1 in the placenta in lines 173 - 178.
Comment 4. Lines 128-137- Authors need to discuss any specifics known on how the placenta affects cancer progression.
Answer: We agree with this comment and have added a new section, “c. Cancer response due to placental factors’’, which specifies some points about how placental factors and hormones can affect cancer cells. This new section is in lines 205 - 268.
Comment 5. The entire section 2.1 provides generalized statements and not many very specific pathways, proteins, mechanisms. The statements are more associative and hint at the knowledge of specific mechanisms being identified in the research. The article should discuss these specifics to give the audience an overview of what is known in the field.
Answer: We agree with this comment. We have now rewritten the manuscript with new subsections to better organise all the ideas, and we have tried to be more specific in pointing out some molecular mechanisms and pathways. As many mechanisms are not completely better understood or elucidated, we have tried to show some points that could give an idea about the way that cancer cells can affect placenta cells, and vice-versa. As suggested in comment #1, this new section was rewritten, and we explored more conceptual points. This new section corresponds to lines 64 -340.
Comment 6. Please describe the model that is referred to in lines 145-153.
Answer: We have rewritten the section and added the animal models described in the text. Most of the experimental models work with rats and mice, and we have the same model in our laboratory. Some studies work with sheep, sows, and BeWo cells in vitro, which we now mention in the main text (lines 287 - 293).
Comment 7. Lines 175-180 do not seem to be related to the preceding statements in the paragraph. What “factors” are the authors referring to in line 175?
Answer: We have rewritten some statements and now focus on not being vague in defining all points, such as the factors mentioned related to the placenta hormones, growth factors, and cytokines released and affecting the cancer cells. We hope this new version does not have any vague points. Please see lines 276 – 768.
Comment 8. Lines 197-215 An expanded explanation is need on immune activity in pregnancy and similar mechanisms in cancer progression/development.
Answer: We have rewritten the statement and tried to discuss more about the mechanisms of immune activity in pregnancy and also cancer cells that avoid the host immune response. These new explanations are in lines 287 - 315
Comment 9. Overall the review addresses a topic of importance given the increasing prevalence of cancer diagnosis during pregnancy. The manuscript reads as if one individual's composed sections 1 and 2 with another individual composed the latter 2 sections. This should be corrected so that the review does not seem so disjointed. Section2 of the manuscript should be revised to add in additional specifics of mechanisms and details regarding what is currently known as was done in section 3. Section 2 seems disjointed and jumps from topic to topic without transition statements to connect the paragraphs and is not reflective of the subheading. Additionally, English type editing is required for the first 2 sections of the manuscript
Answer: We agree with this comment and have tried to change the reviewed manuscript to have more connection among the sections. We also added more description of some mechanisms and pathways to improve this new version.
We have sent the manuscript to American Manuscript Editors to edit the whole manuscript and to better format the manuscript in English style. We hope this new version with corrections is clearer. The certificate of editing is below with the Certificate Verification Key: 068-455-233-121-882

Reviewer 2 Report
I read with great interest the review of de Moraes Santos de Oliveira et al about cellular biology and mechanisms affecting placenta in case of cancer during pregnancy.
The topic is of interest and the manuscript provide a lot of information, sometimes too specialized for the readership of an oncologic journal.
This review is interesting of course.
However, I am disappointed since I was waiting a review more easy-to-read, with a better-structured manuscript.
The article is difficult to understand given the fact that authors are highly (too) specialized in the field. The reader might be lost during the reading of some chapters that appear unclear.
Sometimes there are lacking information (example : general macroscopic findings in women with cancer, in women treated with cancer agents). Moreover, some reminders about placental physiology should be useful since the readers of cancers journal are not specialized in obstetrics.
Please find some comments that should be taken into account:
Perhaps the manuscript should be separated differently. One chapter with a reminder about placental physiology and pathology with clear figures – and another one with the molecular effects?
Specific comments:
L44 and L45: please add some papers published by Frederic Amant’s team, in high-IF reviews.
L64: please correct the typo: “trimester”
L64: please add a reference such Ben Mol et al. Lancet 2016 or Chaiworapongse. Nature Rev Nephrol. 2014.
L78-86 : this chapter is difficult to understand. A clear figure could be useful to shorten the manuscript and ensure that the chapter is clear.
L88: please add a reference.
L104-109: this chapter is not clear and too complex. Please rephrase.
L127: How can we explain the absence of PlGF in other types of cancer? Please explain.
L131: not clear. Please rephrase.
L133-137: which disease? Not clear.
L144: The authors may add information about repercussion of anticancer agents on placental tissues. If yes, please add some data about placental apoptosis. Vereecke and Calsteren published on that topic.
L148: please explain.
L151: “for the fetus”.
L169 and 171: please add some references about Vancalsteren and Amant’s team.
L171: The cognitive development is not impacted by anticancer agents to my knowledge (NEJM Amant et al).
L191: unclear
L215: a figure may be useful in order to make the explanation clearer.
L217: unclear
L221: “exposed to agents that cause oxidative stress » such?….
L226: homograph? Too concise. Please explain.
L226-228: there are reports of fetal metastases. Please explain.
L238: please add a recent reference (such Amant).
L244: are there any data in humans? All these information are “very” preclinical…
L248: in mice ?
L264: in animals? In trophoblast culture? What is “heath pattern”.
L296: reabsorption of what?
L297: MAC?
L305-307: unclear
L310: What is the Warburg effect?
L318: tumoral effects? Please explain.
L320-321: I do not see the link…
It is difficult to know if leucine has an effect in humans?
The conclusion is too firm. Please moderate.
In the Figure: the authors should add a “+” above “the purple arrow”.
The first figure should removed (only one is sufficient).
But another may be useful as a reminder of placental physiology and pathology and/or to explain some of the cellular biology effects of cancer…
Author Response
Campinas, Brazil – March 04th, 2021
Reviewer #2
I read with great interest the review of de Moraes Santos de Oliveira et al about cellular biology and mechanisms affecting placenta in case of cancer during pregnancy.
The topic is of interest and the manuscript provide a lot of information, sometimes too specialized for the readership of an oncologic journal.
This review is interesting of course.
However, I am disappointed since I was waiting for a review more easy-to-read, with a better-structured manuscript.
The article is difficult to understand given the fact that authors are highly (too) specialized in the field. The reader might be lost during the reading of some chapters that appear unclear.
Sometimes there are lacking information (example: general macroscopic findings in women with cancer, in women treated with cancer agents). Moreover, some reminders about placental physiology should be useful since the readers of cancers journal are not specialized in obstetrics.
Answer: We would like to thank you for your comments. We have rewritten the whole manuscript to have better flow and clarity for some points. We tried to add more connection among the sections and added description of some mechanisms and pathways to improve this new version.
Please find some comments that should be taken into account:
Perhaps the manuscript should be separated differently. One chapter with a reminder about placental physiology and pathology with clear figures – and another one with the molecular effects?
Answer: We agree with this point and have sectioned the manuscript in terms of topics. We have rewritten the manuscript and added new subsections to better organise all the ideas, such as the physiology of the placenta and what is altered and worsened in cancer cells. We have tried to be more specific in some points by adding some molecular mechanisms and pathways. Although many mechanisms are not completely understood or elucidated, we have tried to show some points that could give an idea about the way that cancer cells can affect placenta cells, and vice-versa.
Specific comments:
Comment 1. L44 and L45: please add some papers published by Frederic Amant’s team, in high-IF reviews.
Answer: We thank you for the suggestion. We have now added the references: Amant, et al. Management of cancer in pregnancy, Best Pract. Res. Clin. Obstet. Gynaecol. 29 (2015) 741–753. https://doi.org/10.1016/j.bpobgyn.2015.02.006, and Haan, et al. Oncological management and obstetric and neonatal outcomes for women diagnosed with cancer during pregnancy: a 20-year international cohort study of 1170 patients, Lancet Oncol. 19 (2018) 337–346. https://doi.org/10.1016/S1470-2045(18)30059-7 which were included in the main text (lines 38 and 48, respectively).
Comment 2. L64: please correct the typo: “trimester”
Answer: Thanks for pointing out this misspelling. We have now checked the whole manuscript to correct it.
Comment 3. L64: please add a reference such Ben Mol et al. Lancet 2016 or Chaiworapongse. Nature Rev Nephrol. 2014.
Answer: Again, we thank you for this suggestion. We have now included these two papers to improve this review: Chaiworapongsa, et al. Pre-eclampsia part 1: Current understanding of its pathophysiology, Nat. Rev. Nephrol. 10 (2014) 466–480. https://doi.org/10.1038/nrneph.2014.102 and Mol et al.. Pre-eclampsia. Lancet. 2016 Mar 5;387(10022):999-1011. doi: 10.1016/S0140-6736(15)00070-7”. Both are cited in line 55.
Comment 4. L78-86: this chapter is difficult to understand. A clear figure could be useful to shorten the manuscript and ensure that the chapter is clear.
Answer: We agree that a figure would be very useful. We also tried to rewrite the whole manuscript to better clarify the text.
Comment 5. L88: please add a reference.
Answer: The reference was added in line 167
Comment 6. L104-109: this chapter is not clear and too complex. Please rephrase.
Answer: We have rewritten this part of the text while trying to better explain and clarify this point of similarities between trophoblastic and cancer cells. Some studies state that the tumoural characteristics forming metastasis are concerned with the same pattern of trophoblast invasion. Thus, we have rewritten these statements and focus on the mechanisms related to angiogenesis and degradation of the extracellular matrix that are presented in both tumour and trophoblastic cells. This part of the manuscript is now in lines 119 - 138
Comment 7. L127: How can we explain the absence of PlGF in other types of cancer? Please explain.
Answer: We have sectioned the manuscript in terms topics and have rephrased this section. The objective of this section is to highlight the similarities between cancer and trophoblastic cells, and we wrote that PIGF is also higher in many other types of cancer, including breast, bladder, prostate, and colon cancer. This new statement is in lines 191-194
Comment 8. L131: not clear. Please rephrase.
Answer: We agree that this phrase was not clear, so we have subsectioned the manuscript. This point about the epigenetic influence of the placenta inducing foetus programming is included in the whole section about how leucine supplementation can improve the adult offspring of tumour-bearing rats (lines 196 - 199).
Comment 9. L133-137: which disease? Not clear.
Answer: As mentioned above, we agree with this comment. We now mention that the influence of the placenta in foetus programming is improved by leucine supplementation, since this point was much more of a focus in this section (lines 196 - 204).
Comment 10. L144: The authors may add information about the repercussion of anticancer agents on placental tissues. If yes, please add some data about placental apoptosis. Vereecke and Calsteren published on that topic .
Answer: We also agree with this comment, which helped to improve the manuscript. We have included some points about chemotherapy in the context of pregnancy, and also affecting the placenta. Therefore, we have also added this interesting reference in the main text (lines 330 - 334), including that the increased expression of 8-hydroxy-2’-deoxyguanosine (8-OHdG, a marker for oxidative damage to DNA) in trophoblastic cells exposed to chemotherapy for more than 196 days of gestation led to the restriction of foetus growth.
Comment 11. L148: please explain.
Answer: As mentioned, we have rewritten and rephrased the main parts of the whole manuscript, including this line. The initial text had points about our previous results showing deep foetal weight loss as well as the foetal growth in the experimental model of cancer cachexia indicating intense damage for both mother and foetus in the rat pregnancy“ (lines 270 - 275)
Comment 12. L151: “for the fetus”.
Answer: The paragraph was rewritten, and we tried to be clearer, as the lower glucose uptake occurred for the host in the pregnant tumour-bearing group. Unfortunately, we did not measure this parameter in the foetus, but we are now using an experimental procedure to better understand the dynamic transport of glucose and other nutrients in cancer/pregnancy conditions.
Comment 13. L169 and 171: please add some references for Vancalsteren and Amant’s team.
Comment 14. L171: The cognitive development is not impacted by anticancer agents to my knowledge (NEJM Amant et al).
Answer to the comments 13 and 14: We agree with these comments. We found recent references from Amant about the effects of cancer drugs during pregnancy in foetal cognitive development, which oppose our previous reference written in the review. Thus, we rewrote this paragraph while adding some references from Amant’s team, including some information of chemotherapy effects in the placenta without affecting the foetus. These new statements are in the section, “d. Placental impairment due to cancer association”, in lines 327 - 334
Comment 15. L191: unclear
Answer: As preeclampsia is not the focus of this review, this information was excluded when rewriting this paragraph.
Comment 16. L215: a figure may be useful to make the explanation clearer.
Answer: We agree with this comment. We added more information and hope that this section is better explained and clearer (lines 294 - 306).
Comment 17: L217: unclear / Comment 18. L221: “exposed to agents that cause oxidative stress » such?…
Answer: We have rephrased this paragraph and subsectioned the text. We included this information in the new section, “d. Placental impairment due to cancer association”, showing that in assays in vitro, the proteolysis-inducing factor produced by cancer cells (in this case by the Walker 256 tumour) could increase the damage to DNA and oxidative stress in trophoblastic cells (lines 291 - 293).
Comment 19. L226: homograph? Too concise. Please explain. / Comment 20. L226-228: there are reports of fatal metastases. Please explain
Answer: We have rearranged the whole manuscript and we excluded the word “homograph”. This paragraph was rephrased to better explain how a foetus’s immune system could avoid metastasis, pointing to some mechanism related to the immune system in the mother, which could allow the metastasis process. We tried to better explain this in lines 320 - 324.
Comment 21. L238: please add a recent reference (such Amant).
Answer: We have now included the reference from Amant’s team to improve the information in this paragraph (lines 38 and 48).
Comment 22. L244: are there any data in humans? All these information are “very” preclinical…
Answer: The data about leucine supplementation in the context of cancer and pregnancy is already being studied in animals and cell cultures. However, the use of leucine as a dietary supplementation is already approved for some cases, as explained in lines 350 - 353
Comment 23. L248: in mice?
Answer: The paragraph was rewritten, and we included that the information about the experimental model using mice (line 372).
Comment 24. L264: in animals? In trophoblast culture? What is “heath pattern”.
Answer: We have rephrased this statement and included the information that leucine guarantees mTOR pathway maintenance, which is essential for cellular growth and proliferation in skeletal muscle, the uterus, and placenta tissue in experimental model with rats.
Comment 25. L296: reabsorption of what?
Answer: The word “reabsorption” was corrected to “foetal resorption” (line 373).
Comment 26. L297: MAC?
Answer: We have now explained in the text that an experimental model of cancer-induced cachexia named murine colon adenocarcinoma (MAC16) is actually used in our experiments associated with pregnancy (lines 372).
Comment 27. L305-307: unclear
Comment 28. L310: What is the Warburg effect?
Answer: We have rephrased the paragraph, and the explanation for the Warburg effect was included in lines 105 - 110.
Comment 29. L318: tumoral effects? Please explain.
Answer: This paragraph was rewritten to be clearer. We intended to say the “tumoural damage effects” (lines 387 - 388).
Comment 30. L320-321: I do not see the link…
Answer: As the main text was rearranged and restructured, this information was excluded, and a new paragraph was written to finalize this section (lines 386 - 393).
Comment 31. It is difficult to know if leucine affects humans?
Answer: It is known that leucine has effects in humans in some cases (we have added a few lines pointing out this information (lines 350 - 353), but the effects in the context of cancer and pregnancy are already being studied).
Comment 32. The conclusion is too firm. Please moderate.
Answer: The conclusion was rewritten in a moderate way (lines 396 - 402).
Comment 33. In the Figure: the authors should add a “+” above “the purple arrow”.
Answer: We have added a question tag and also +/-, as we have seen that some hormones increase the activity of mTOR in cancer cells, but others can inhibit it.
Comment 34. The first figure should be removed (only one is sufficient).
Answer: We agree with this comment, and now show all points added to the leucine effect.
Comment 35. But another may be useful as a reminder of placental physiology and pathology and/or to explain some of the cellular biology effects of cancer…
Answer: As mentioned above, we rewrote the manuscript and divided it into sections, including the section, “a. Placental physiology” (line 65 - 97), as well as other subsections about how the placenta factors/hormones could affect cancer cells, and also a section reporting cancer damage effects in the placenta tissue (lines 205/ and 269).
We have sent the manuscript to American Manuscript Editors to edit the whole manuscript and to better format the manuscript in English style. We hope this new version with corrections is clearer. The certificate of editing is below with the Certificate Verification Key: 068-455-233-121-882

Reviewer 3 Report
This is an interesting review manuscript discussing the similarities between the placenta/pregnancy and cancer biology and the importance of further investigation. This is a systematic review.
1. Could the authors supply a supplementary file on their search strategy? Describe all information sources including dates and other restrictions (if any) in the search and date searched? Present the full electronic search strategy so it can be repeated? How were studies selected? Please provide a flow diagram of how studies were selected? How was the data obtained from the studies? Was the data recorded on a piloted form? Was the data recorded in an electronic database? What was done to ensure accuracy of data transfer? What was done if missing data?
2. As the placenta is a very hormonally active organ could the authors discuss the potential role of placental hormone production and its role in cancer?
3. In the abstract the authors note "postponing family planning." I think most would suggest that they are using family planning to delay pregnancy until later in life. Please clarify?
4. There are a number of grammatical/syntax errors making it difficult to follow. I would suggest submission to an English editor with experience in scientific publication prior to resubmission.
Author Response
Campinas, Brazil – March 04th, 2021
Reviewer #3
This is an interesting review manuscript discussing the similarities between the placenta/pregnancy and cancer biology and the importance of the further investigation. This is a systematic review.
Answer: We would like to thank for the time spent in analysing our work and all suggestions, which contributed to improving our work. We also would like to thank you for the compliment about the value of the review. However, we would like to clarify that our review is not a systematic review but a narrative review.
Comment 1: Could the authors supply a supplementary file on their search strategy? Describe all information sources including dates and other restrictions (if any) in the search and date searched? Present the full electronic search strategy so it can be repeated? How were the studies selected? Please provide a flow diagram of how studies were selected? How was the data obtained from the studies? Was the data recorded on a piloted form? Was the data recorded in an electronic database? What was done to ensure the accuracy of data transfer? What was done if missing data?
Answer: Unfortunately, this work is not a systematic review. We wrote a descriptive narrative review, so we cannot have a search strategy and all related data to present here. We have mentioned in the objective of this review that “the present article aimed to describe a narrative review on the relationship of placental effects in cancer cells, and vice-versa, as well as how leucine participates in modulating their interaction” (lines 61- 63).
Comment 2. As the placenta is a very hormonally active organ could the authors discuss the potential role of placental hormone production and its role in cancer?
Answer: We thank you for raising this point. We have now sectioned the manuscript, and one of the sections is about the placenta physiology (lines 65), and another is about some effects of the placental factors/hormones in cancer cells (lines 205).
Comment 3. In the abstract, the authors note "postponing family planning." I think most would suggest that they are using family planning to delay pregnancy until later in life. Please clarify?
Answer: We would like to apologise for this, and we agree that women are postponing pregnancy through family planning. We have now corrected this statement in the abstract section and in the introduction (lines 23).
Comment 4. There are several grammatical/syntax errors making it difficult to follow. I would suggest submission to an English editor with experience in a scientific publication before resubmission.
Answer: We have sent the manuscript to American Manuscript Editors to edit the whole manuscript and to better format the manuscript in English style. We hope this new version with corrections is clearer. The certificate of editing is below with the Certificate Verification Key: 068-455-233-121-882
